# Efficient Heterogeneity-Aware Federated Active Data Selection

**Ying-Peng Tang** [1]  **Chao Ren** [2] [1]  **Xiaoli Tang** [1]  **Sheng-Jun Huang** [3]  **Lizhen Cui** [4]  **Han Yu** [1]

## Abstract

Federated Active Learning (FAL) aims to learn an effective global model, while minimizing label queries. Owing to privacy requirements, it is challenging to design effective active data selection schemes due to the lack of cross-client query information. In this paper, we bridge this important gap by proposing the Federated Active data selection by LEverage score sampling (`FALE`) method. It is designed for regression tasks in the presence of non-i.i.d. client data to enable the server to select data globally in a privacy-preserving manner. Based on FedSVD, `FALE` aims to estimate the utility of unlabeled data and perform data selection via leverage score sampling. Besides, a secure model learning framework is designed for federated regression tasks to exploit supervision. `FALE` can operate without requiring an initial labeled set and select the instances in a single pass, significantly reducing communication overhead. Theoretical analyze establishes the query complexity for `FALE` to achieve constant factor approximation and relative error approximation. Extensive experiments on 11 benchmark datasets demonstrate significant improvements of `FALE` over existing state-of-the-art methods.

## 1. Introduction

Training effective machine learning models typically relies on large-scale labeled datasets, which are often expensive and time-consuming to obtain. Active Learning (AL) (Settles, 2009) addresses this issue by selectively querying the most useful unlabeled data points for improving the model

[1]College of Computing and Data Science, Nanyang Technological University, Singapore [2]School of Electrical Engineering and Computer Science, KTH Royal Institute of Technology, Sweden [3]College of Computer Science and Technology, Nanjing University of Aeronautics and Astronautics, Nanjing, China [4]School of Software, Shandong University, Jinan, China. Correspondence to: Han Yu <han.yu@ntu.edu.sg>.

*Proceedings of the $42^{nd}$ International Conference on Machine Learning*, Vancouver, Canada. PMLR 267, 2025. Copyright 2025 by the author(s).

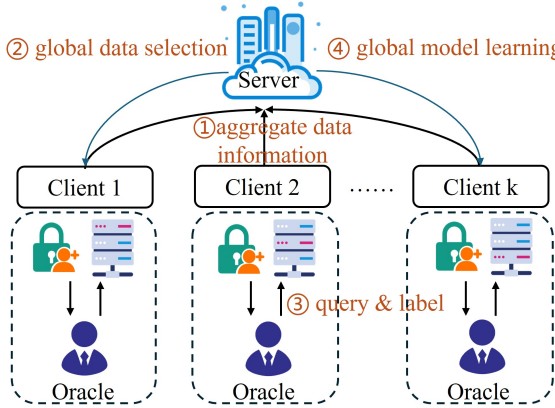

*Figure 1.* The overall framework of the proposed method. First, the server aggregates data information from each client to facilitate data selection. Next, it selects the most informative data points and returns their corresponding indexes to the respective clients. The clients then label the selected data locally. Finally, the global model is trained on the server using the masked features and labels of the queried data.

performance from an oracle for labeling, thereby reducing the labeling cost while maintaining high model performance. Traditional AL approaches usually focus on the centralized learning setting, where all the data is readily accessible. However, in many real-world applications, data is stored in a distributed manner, raising the need to consider communication costs, or across multiple parties, in which the data cannot be directly accessed due to privacy concerns and data governance regulations (Regulation, 2016). In these challenging settings, existing methods often do not sufficiently address data privacy, computational constraints, and communication overhead. Consequently, their performance can be limited.

Federated Learning (FL) (Yang et al., 2019; 2020) emerges as an effective learning paradigm to address these challenges by enabling collaborative model training across multiple decentralized devices or institutions. It tries to learn an accurate global model by aggregating local model updates from multiple clients, without sharing raw data. Due to the high demand for data privacy and security, each client usually needs to label their own data locally and only share the model updates during the learning process (Yang, 2021;

Ren et al., 2024; Weng et al., 2025). However, this yields a high risk of knowledge overlapping among clients data, thus waste of labeling costs. This necessitates the research on Federated Active Learning (FAL) (Wu et al., 2022; Kim et al., 2023; Zhang et al., 2023a; Cao et al., 2023), to enable label-efficient and privacy-preserving model learning in decentralized environments.

In FAL, each client possesses its own dataset, which may contain limited or no initially labeled data, and the data distribution can vary significantly across clients. The objective is to learn an effective global model while minimizing the number of data queries among clients, all under strict privacy constraints. The major challenges in FAL includes 1) designing a data selection algorithm that operates without direct access to the data and within a limited communication budget, and 2) establishing theoretical guarantees of the query complexity of the active selection algorithm in distributed and privacy-preserving learning scenario. Existing FAL methods typically employ local data selection strategies to comply with the privacy regulation, where each client independently selects data samples for labeling based on local model (Wu et al., 2022), global model (Kim et al., 2023) and data distributions (Zhang et al., 2023b). These approaches can unavoidably result in overlapping queries due to the limited coordination among clients and incomplete knowledge of the global data landscape. Furthermore, the theoretical guarantees regarding the query complexity of active selection algorithms remain underexplored.

In this paper, we propose a novel FAL method, called FALE, for regression task on non-i.i.d. client data, as shown in Fig. 1. Our proposed method designs a global selector that minimizes redundant information across clients, as well as a global model learning paradigm, supported by theoretical guarantees on the query complexity. To achieve the effective data selection, our method leverages the latest FedSVD method (Chai et al., 2022; 2024), which is a privacy-preserving method for Singular Value Decomposition (SVD) in FL setting. FedSVD provides an efficient mechanism to gathering global information without exposing individual client data. Based on these results, we employ leverage score sampling (Mahoney, 2011; Woodruff et al., 2014) to identify and query the most informative data points. To fully exploit the queried supervision, we further propose a model learning paradigm for federated regression. It securely trains a global model on the server by using masked feature matrices and label vectors of the queried data. Then, the learned model can be unmasked with a secret matrix on the client side. In this way, FALE can operate with no initial labeled set and select instances in a single pass, substantially reducing communication overhead. Furthermore, our theoretical analysis shows that our proposed FALE method achieves a constant factor approximation using $\mathcal{O}(d \log d)$ queries, and relative error approximation

using $\Omega(d \log d + d/\epsilon)$ queries with high probability in FAL setting, where $d$ is the feature dimension, $\epsilon$ is the error control parameter. We demonstrate the effectiveness of our proposed FALE approach through extensive experiments on 11 benchmark datasets, showing significant improvements over existing methods.

## 2. Related Works

AL (Settles, 2009; Ren et al., 2021) focuses on selective querying of high-potential unlabeled data. Numerous query strategies have been developed, generally falling into two categories: informativeness-based (Kirsch et al., 2019; Huang et al., 2024) and representativeness-based (Sener & Savarese, 2018; Sinha et al., 2019) approaches. Informativeness-based methods measure the prediction uncertainties of data points, while representativeness-based methods aim to capture the underlying data distribution by data selection. However, most existing AL algorithms assume centralized data storage, which restricts their applicability in decentralized settings.

Leverage score sampling is an effective technique with provable guarantees for active $\ell_2$-regression problems $\min_{\boldsymbol{\theta}} \|X\boldsymbol{\theta} - \boldsymbol{y}\|_2$ (Woodruff et al., 2014; Mahoney, 2011), where the feature matrix $X \in \mathbb{R}^{n \times d}$ is known while the label vector $\boldsymbol{y} \in \mathbb{R}^n$ needs to be queried with some costs. Theoretical analyses are widely conducted for obtaining $(1 + \epsilon)$-approximate solutions, i.e., $\|X\boldsymbol{\theta}' - \boldsymbol{y}\|_2 \leq (1 + \epsilon)\|X\boldsymbol{\theta}^* - \boldsymbol{y}\|_2$, where $\boldsymbol{\theta}'$ is the output of the algorithm and $\boldsymbol{\theta}^*$ is the true minimizer. For examples, Mahoney (2011) show that $\mathcal{O}(d \log d + d/\epsilon)$ suffices to solve. Chen & Price (2019) propose an algorithm that achieves the optimal $\mathcal{O}(d/\epsilon)$ query complexity. Besides, the time complexities have also been analyzed (Woodruff et al., 2014). However, existing methods are designed for centralized learning settings and its applicability to privacy-preserving and distributed learning scenarios is underexplored.

FL (Liu et al., 2024; Ren et al., 2025; Fan et al., 2025) is a decentralized approach to machine learning where multiple clients collaboratively train a model without sharing their raw data. Perhaps the most popular setting is the horizontal FL (Yang et al., 2019), where multiple clients have the same feature space but different data distributions. Furthermore, according to the data distributions among clients, it can be further categorized into i.i.d. (Yang, 2021) and non-i.i.d. (Lu et al., 2024) settings. Most of existing FAL methods focus on i.i.d. setting and apply off-the-shelf AL algorithms on each local node independently (Wu et al., 2022; Zhang et al., 2023b). The selections from different clients may contain repeated and redundant supervision, leading to suboptimal global model performance. There are some works studying the non-i.i.d. client data for federated active classification problem (Kim et al., 2023; Zhang et al., 2023a; Cao et al.,

2023). They share a similar idea that considers both informativeness and representativeness in data selection. However, the querying is also conducted locally, which can be myopic and suboptimal. Besides, the theoretical guarantees of the query complexity has not been analyzed.

# 3. Preliminaries

**Notations and Problem Settings** Throughout the paper, we denote scalars by lowercase letters, vectors by bold lowercase letters, and matrices by bold uppercase letters. $\|\cdot\|$ denotes the norm of a vector. $e_j$ represents the $j$-th standard basis vector, where the $j$-th entry is equal to 1, and all other entries are 0. This work focuses on the horizontal FL with non-i.i.d. data, where multiple clients have the same feature space but different data distributions. Assuming that there are $k$ clients, each client $i$ has its own dataset $\mathcal{D}_i = \{x_i^j\}_{j=1}^{n_i}$, which is assumed to be fully unlabeled. $x_i^j \in \mathbb{R}^d$ is the feature vector. We denote $X_i = [x_i^1, \ldots, x_i^{n_i}]^\top$ as the feature matrix of data in client $i$, and $y_i = [y_i^1, \ldots, y_i^{n_i}]^\top \in \mathbb{R}^{n_i}$ is the unknown label vector. $n_i$ is the number of data points in client $i$, and $n = \sum_{i=1}^k n_i$. Denote by $X = [X_1^\top, \ldots, X_k^\top]^\top \in \mathbb{R}^{n \times d}$. We assume that $X$ is column full-rank and $n \gg d$.

Initially, each client has an empty labeled dataset $\mathcal{L}_i = \emptyset$ and an unlabeled set $\mathcal{U}_i = \mathcal{D}_i$. At each FAL iteration, each client $i$ selects a subset of data $\mathcal{Q}_i \subset \mathcal{U}_i$ for querying and update the sets $\mathcal{L}_i = \mathcal{L}_i \cup \mathcal{Q}_i$, $\mathcal{U}_i = \mathcal{U}_i \setminus \mathcal{Q}_i$. Then, a global model $\theta^g$ is trained using the queried data by, e.g., FedAvg (McMahan et al., 2017). Existing FAL methods iteratively repeat this cycle until the querying budget is exhausted or specific performance criteria are met. In contrast, the method proposed in this paper is a one-pass querying approach, selecting all data points in a single iteration.

**FedSVD** FedSVD (Chai et al., 2022) calculates the SVD of the entire dataset securely in FL setting. The main steps of the FedSVD is summarized as follows

(1) A trusted authority generates random orthogonal matrices $P \in \mathbb{R}^{n \times n}$ and $Q \in \mathbb{R}^{d \times d}$, where $P$ can be simplified as $\mathrm{diag}(P_1, \ldots, P_k)$ for computational efficiency, $P_i \in \mathbb{R}^{n_i \times n_i}$ is a random orthogonal matrix. $Q$ is sent to all clients and $P_i$ is sent to client $i$.
(2) All clients mask their local data by $X_i' = P_i X_i Q$ and send $X_i'$ to the FL server.
(3) The FL server securely aggregates $X'$ from all clients.
(4) The FL server performs SVD to obtain $X' = U' \Sigma V'^\top$, where $U' \in \mathbb{R}^{n \times n}$, $\Sigma \in \mathbb{R}^{n \times d}$, and $V' \in \mathbb{R}^{d \times d}$.
(5) Users download the decomposed matrices and recover them using $P_i$ and $Q$. After recovering, each user will have the complete $\Sigma$ and $V^\top$, but only part of $U$, denoted by $U_i$, that corresponds to their local data.

**Leverage Score Sampling** Leverage score sampling (Mahoney, 2011; Woodruff et al., 2014) samples rows in a matrix with probability proportional to their leverage scores:

**Definition 3.1** (Statistical Leverage Score). The leverage score $\tau^i(X)$ of the $i^\text{th}$ row $x_i$ of a matrix $X \in \mathbb{R}^{n \times d}$ is equal to:

$$\tau^i(X) = x_i^T \left(X^T X\right)^{-1} x_i \qquad (1)$$

Moreover, when $X$ is column full-rank, $\tau^i(X) = \|e_i^\top U\|^2$, where the columns of $U$ is an orthogonal basis of the column space of $X$.

**Importance Sampling Matrix** Importance sampling adjusts the weights of sampled data points to preserve the underlying distribution.

**Definition 3.2** (Importance Sampling Matrix). Let $\{p_1, \ldots, p_n\} \in (0, 1]^n$ be a given set of probabilities with $\sum_i p_i = 1$. A matrix $S$ is an $n_s \times n$ importance sampling matrix if each of its rows is chosen to equal $\frac{1}{\sqrt{n_s \cdot p_i}} \cdot e_i$ with probability proportional to $p_i$. $n_s$ is the sampling size.

# 4. The Proposed FALE Method

## 4.1. Global Data Selection

To perform global data selection in FL, we need to gathering necessary information of all clients' data, while preserving privacy. Fortunately, a recent technique called FedSVD (Chai et al., 2022) provides a secure and efficient way to achieve this goal. As introduced in Sec. 3, it calculates the SVD results for all clients' data, which provides abundant information for data selection. Therefore, our proposed FALE method first invokes FedSVD as a subroutine. Notably, the clients are only required to recover the left singular matrix to reduce the communication cost. After this process, each client will have $U_i$. Next, each client calculates the leverage scores of their local data $\tau_i = [\tau_i^1, \ldots, \tau_i^{n_i}]$, where $\tau_i^j = \|e_j^\top U_i\|^2$. The clients then send the leverage scores to the FL server for aggregation.

On the server side, it concatenates the leverage scores to obtain $\tau = [\tau_1, \ldots, \tau_k]$, and calculates the sampling probability $p = [p_1, \ldots, p_n]$, where $p_i = \tau^i / \|\tau\|_1$. Given the query budget $n_q$, the server performs i.i.d. sampling of integers from the set $[1, \ldots, n]$ with probability $p$. Note that duplicates may appear during sampling, they only affect the training weights and do not reduce the query budget. The server continues sampling until $n_q$ distinct data points have been selected, and we denote the total number of samples drawn as $n_s$. Denote by $\mathcal{Q} = \{q_1, \ldots, q_{n_s}\}$ the set of sampled integer indices. Construct an importance sampling matrix $S \in \mathbb{R}^{n_s \times n}$, where each row $i$ is equal to $\frac{1}{\sqrt{n_s p_{q_i}}} \cdot e_{q_i}$ with probability $p_i$, for $i = 1, \ldots, n_s$. After sampling, the server sends the selected indices to the respective clients, which label these data points locally.

## 4.2. Global Model Learning

One simple way to learn the global model is to perform a global aggregation of the local models from all clients, e.g., using FedAvg (McMahan et al., 2017). However, this method can be suboptimal due to the non-i.i.d. data distribution among clients. Inspired by Chai et al. (2022), we propose to learn a global model directly on the server using the masked feature matrix and label vector of the queried data points.

Specifically, denote by $X_i^{\mathcal{L}} \in \mathbb{R}^{n_i^l \times d}$ and $\boldsymbol{y}_i^{\mathcal{L}} \in \mathbb{R}^{n_i^l}$ as the feature matrix and label vector of the labeled data in client $i$, where $n_i^l$ is the number of labeled data points. Clients mask their feature matrix by $P_i X_i^{\mathcal{L}} Q$ and the label vector by $P_i \boldsymbol{y}_i$, where $Q \in \mathbb{R}^{d \times d}$ is the orthogonal matrix generated in FedSVD, $P_i \in R^{n_i^l \times n_i^l}$ is an arbitrary random orthogonal matrix generated by client $i$ locally. The masked feature matrix and label vector are then sent to the server for aggregation. According to Chai et al. (2022), such a masking operation can preserve the privacy of individual data, as given a masked feature matrix $X' = [PXQ]$, there are infinitely many possible $\bar{X}$ can be masked into $X'$.

The server performs secure aggregation, as Step 3 in FedSVD, to obtain the complete masked feature matrix $[PX^{\mathcal{L}}Q]$ and the label vector $P\boldsymbol{y}^{\mathcal{L}}$, where $X^{\mathcal{L}} = [X_1^{\mathcal{L}}, \ldots, X_k^{\mathcal{L}}]$ and $\boldsymbol{y}^{\mathcal{L}} = [\boldsymbol{y}_1^{\mathcal{L}}, \ldots, \boldsymbol{y}_k^{\mathcal{L}}]$, $P = \text{diag}(P_1, \ldots, P_k)$. The importance sampling matrix $S$ is then transformed to a reweighting matrix such that the non-zero element in each row is moved to the diagonal, denoted by $S' \in \mathbb{R}^{n_s \times n_s}$. The server solves the following optimization problem to learn the global model:

$$\min_{\boldsymbol{\theta}} \|[PS'X^{\mathcal{L}}Q]\boldsymbol{\theta} - PS'\boldsymbol{y}^{\mathcal{L}}\|_2^2, \tag{2}$$

or equivalently,

$$\min_{\boldsymbol{\theta}} \|[PSXQ]\boldsymbol{\theta} - PS\boldsymbol{y}\|_2^2. \tag{3}$$

Once the server computes a solution $\hat{\boldsymbol{\theta}}^g$ to the optimization problem (2), it sends $\hat{\boldsymbol{\theta}}^g$ to all clients. Each client then recovers the prediction model by computing $\boldsymbol{\theta}^g = Q^\top \hat{\boldsymbol{\theta}}^g$.

Our proposed global model learning method offers several benefits. First, it preserves the privacy of raw data and the prediction model, as the global model learned on the server needs to be recovered on the client side. Second, solving the optimization problem with masked data is equivalent to the original regression problem, thereby fully utilizing the available supervisory information.

Overall, our proposed FALE is summarized in Algorithm 1.

---

**Algorithm 1** The FALE Algorithm

**Input:** query budget $n_q$, feature matrices $X_i, i = 1, \ldots, k$.
**Output:** prediction model $\boldsymbol{\theta}^g$
1: **Initialization:** $n_s \leftarrow 0$, $\boldsymbol{s} \leftarrow [\,]$, $\mathcal{Q}_i \leftarrow \emptyset$, for $i = 1, \ldots, k$.
   *Global Data Selection:*
2: Perform FedSVD and each client obtains $U_i$, the left singular matrix that corresponds to their local data
3: Each client $i$ calculates $\boldsymbol{\tau}_i^j = \|\boldsymbol{e}_j^\top U_i\|^2$ and sends to the server
4: Server aggregates the leverage scores $\boldsymbol{\tau} = [\boldsymbol{\tau}_1, \ldots, \boldsymbol{\tau}_k]$ and calculates sampling probabilities $\boldsymbol{p}$, where $p_j = \boldsymbol{\tau}^j / \|\boldsymbol{\tau}\|_1$
5: **while** Fewer than $n_q$ distinct elements sampled **do**
6:     $q \leftarrow$ sample a number from $\{1, \ldots, \text{len}(\boldsymbol{\tau})\}$ with replacement with probability $\boldsymbol{p}$
7:     $n_s \leftarrow n_s + 1$
8:     $\boldsymbol{s} \leftarrow \text{append}(1/\sqrt{p_q})$
9:     $i \leftarrow q$ belongs to which client
10:     append $q$ to $\mathcal{Q}_i$
11: **end while**
12: $\boldsymbol{s}_j \leftarrow \boldsymbol{s}_j / \sqrt{n_s}$ for $j = 1, \ldots, n_s$
13: Server sends $\mathcal{Q}_1, \ldots, \mathcal{Q}_k$ to corresponding clients
   *Global Model Learning:*
14: Each client $i$ locally labels the selected data points in $\mathcal{Q}_i$. {*Note that repeated entries in $\mathcal{Q}_i$ will not use extra querying budget*}
15: Clients mask $X_i^{\mathcal{L}}$ and $\boldsymbol{y}_i^{\mathcal{L}}$ by $P_i X_i^{\mathcal{L}} Q$ and $P_i \boldsymbol{y}_i^{\mathcal{L}}$ and send to the server
16: $S' \leftarrow \text{diag}(\boldsymbol{s})$
17: Server aggregates masked data and solves $\hat{\boldsymbol{\theta}}^g = \arg\min_{\boldsymbol{\theta}} \|[PS'X^{\mathcal{L}}Q]\boldsymbol{\theta} - PS'\boldsymbol{y}^{\mathcal{L}}\|_2^2$
18: Server sends $\hat{\boldsymbol{\theta}}^g$ to clients
19: Clients recover the prediction model $\boldsymbol{\theta}^g = Q^\top \hat{\boldsymbol{\theta}}^g$
20: **Return:** $\boldsymbol{\theta}^g$

---

## 4.3. Analysis

### 4.3.1. PRIVACY ANALYSIS

**The server cannot reveal the original matrix and prediction model:** The privacy of the raw data is guaranteed by the FedSVD algorithm, as the masked feature matrix $X'$ can be mapped to infinitely many raw feature matrices $\bar{X}$. In the selection phase, the server only has information of the $\ell_2$ norm of the left singular value matrix of local client data. After obtaining the selection probability, the server distributes the index of selected data to each client for labeling, it cannot recover the raw data. The privacy of the prediction model is also guaranteed with a similar argument, as the global model learned on the server is encrypted by a secret random orthogonal matrix $Q$. According to the Theorem 5 in (Zhang et al., 2019), the masked vector is

computationally indistinguishable from a random vector.

**The users cannot know the data owned by the others:** In our proposed FALE method, each client holds local data and $U_i$, $P_i$ that correspond to their data. Although all clients share the same matrix $Q$, knowing $Q$, $U_i$, $P_i$ is insufficient to reconstruct any other client's local data, as discussed in Chai et al. (2022). During the model learning phase, the global model trained on selected instances may reflect a coarse distribution of the global dataset, but it does not directly reveal the underlying data.

### 4.3.2. COMMUNICATION COST ANALYSIS

The overall communication cost of FALE includes four main components: (1) performing FedSVD, (2) transmitting leverage scores and the indices of selected data, (3) uploading masked feature matrices and label vectors, and (4) sharing the global model.

FedSVD is highly efficient and scalable to billion-scale datasets. For a detailed analysis of its communication overhead, we refer to the work (Chai et al., 2022). The transmission of leverage scores and selected data indices incurs relatively low costs, with complexities of $\mathcal{O}(n)$ and $\mathcal{O}(n_s)$, respectively. Since the masked feature matrices retain the same dimensionality as the raw data, their transmission does not introduce significant additional overhead. Furthermore, the communication cost associated with global model transmission remains consistent with standard FL frameworks.

A key advantage of FALE is its single-pass selection approach, where communication is performed only once. This makes it a communication-efficient solution.

### 4.3.3. THEORETICAL ANALYSIS

We analyze the query complexity of FALE as follows.

**Theorem 4.1.** *Considering federated learning with non-i.i.d. data. Let $k$ be the number of clients, $\epsilon \in (0, 1]$ be an error parameter, $X_i \in \mathbb{R}^{n_i \times d}$ be the corresponding data matrices and $\boldsymbol{y}_i \in \mathbb{R}^{n_i}$ be the initially unknown target vector. Denote by $X = [X_1^\top, \ldots, X_k^\top]^\top$ and $\boldsymbol{y} = [\boldsymbol{y}_1^\top, \ldots, \boldsymbol{y}_k^\top]$, $\boldsymbol{\theta}^* = \arg\min_{\boldsymbol{\theta}} \|X\boldsymbol{\theta} - \boldsymbol{y}\|_2^2$. Algorithm 1 computes the global leverage scores for the data in each client. Moreover, if Algorithm 1 queries $\mathcal{O}(d \log d)$ data points and outputs the model $\boldsymbol{\theta}^g$, it holds with probability at least 0.99 that*

$$\|X\boldsymbol{\theta}^g - \boldsymbol{y}\|_2^2 \leq \alpha \|X\boldsymbol{\theta}^* - \boldsymbol{y}\|_2^2, \tag{4}$$

*with constant $\alpha$. In addition, if it queries $\Omega(d \log d + d/\epsilon)$ data points and outputs $\boldsymbol{\theta}^g$, it holds with probability at least $1/3$ that*

$$\|X\boldsymbol{\theta}^g - \boldsymbol{y}\|_2^2 \leq (1 + \epsilon)\|X\boldsymbol{\theta}^* - \boldsymbol{y}\|_2^2. \tag{5}$$

**Remark:** Our proposed algorithm selects unlabeled data proportional to their leverage scores and re-weights them in global training phase. Using the properties of leverage score sampling, this approach provides a subspace embedding (Woodruff et al., 2014) of $X$, which is a key component in proving our theorem. The detailed proof is provided in the appendix A.

Our theoretical results establish that the query complexity of our method in the FAL setting matches with the leverage score sampling of the centralized setting (Mahoney, 2011; Derezinski et al., 2018). By securely computing the leverage scores across different clients in the FL framework, we establish that leverage score sampling can be effectively applied to a privacy-preserving decentralized learning environment without any degradation in performance.

## 5. Experiments

### 5.1. Experimental Setup

We employ 9 UCI (Dua & Graff, 2017) and OpenML (Bischl et al., 2021) regression benchmarks in our experiments. The information of is summarized in Table 1. For each dataset, we uniformly sample 20% of the instances for testing, while the remaining instances are distributed across $k = 10$ clients in a non-i.i.d. manner, with each client receiving a different number of instances. To simulate the non-i.i.d. setting in regression task, we perform binning on the regression target vector, with the number of bins equal to the number of clients. We then adopt the Dirichlet distribution strategy (Yurochkin et al., 2019) with a Dirichlet alpha of 5 to assign instances to clients using the bins as a class label. Note that some clients may not have instances from certain bins. We believe this is a common scenario in real-world applications. Since most existing FAL methods require an initial labeled set, we randomly select 1% of the instances from each client to form the initial labeled set. FALE does not require any initial labeled data, although it can accommodate an initial labeled set if available. At each iteration, we allocate a query budget of 5 instances per client, resulting in a total of 50 instances queried per round. Our method performs

*Table 1.* The summary of datasets.

| Dataset (Reference) | # Ins. | # Fea. |
| --- | --- | --- |
| ct (Graf et al., 2011) | 53500 | 379 |
| kegg_undir (Shannon et al., 2003) | 64608 | 27 |
| online_video (Deneke et al., 2014) | 68784 | 26 |
| wec_sydney (Neshat et al., 2018) | 72000 | 48 |
| sarcos (Vijayakumar & Schaal, 2000) | 44484 | 21 |
| diamonds (OpenML ID: 42225) | 53940 | 29 |
| stock (OpenML ID: 1200) | 59049 | 9 |
| protein (OpenML ID: 42903) | 45730 | 9 |
| mlr_knn_rng (OpenML ID: 42454) | 111753 | 132 |

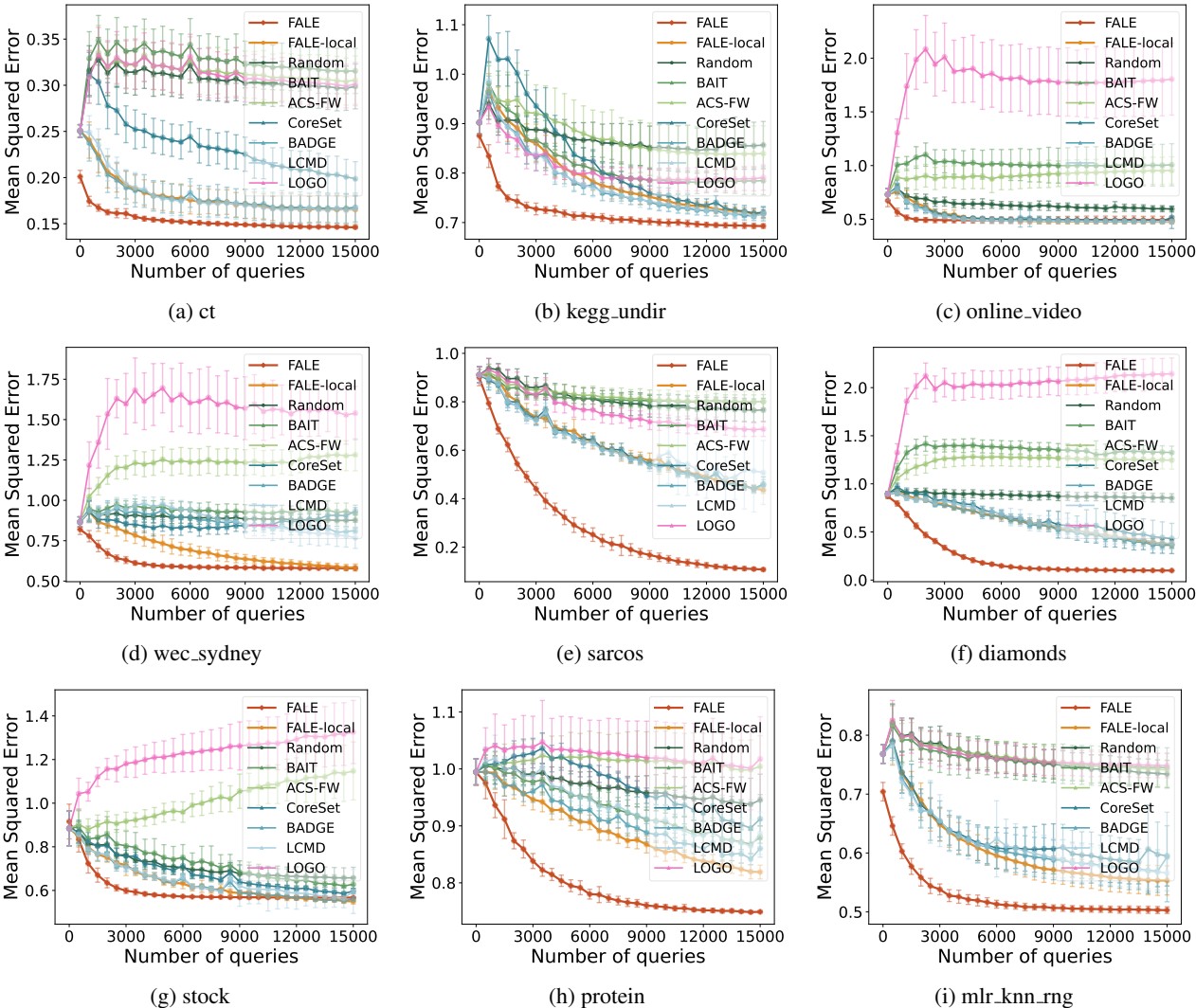

*Figure 2.* The learning curves of the compared methods. The error bars represent the standard deviation of the performances over 10 runs.

global data selection within this same budget. The data splitting process is repeated 10 times, and the performance comparison results are averaged over these runs.

For comparison, we include the state-of-the-art FAL method LOGO (Kim et al., 2023) in our experiments. Although LOGO is originally designed for classification tasks, Kim et al. (2023) suggests that its uncertainty scoring function can be replaced with alternative informativeness scoring functions. Based on this, we extend LOGO to the regression setting by replacing its entropy-based selection mechanism with the regression-specific uncertainty metric BAIT (Ash et al., 2021). Other existing FAL methods are primarily designed for classification tasks, and adapting them to regression problems remains a non-trivial challenge. In addition, we compare our approach with state-of-the-art active regression methods that operate locally within each client. Specifically, the following methods are included:

- **Random:** Uniformly select unlabeled data points for querying.
- **BAIT** (Ash et al., 2021): Select informative data points by optimizing a bound on the maximum likelihood estimator error in terms of the Fisher information.
- **ACS-FW** (Pinsler et al., 2019): Select a batch of representative data by optimizing a sparse subset approximation to the log posterior induced by the full dataset.
- **CoreSet** (Sener & Savarese, 2018): Select a batch of representative data points that minimizes the maximum distance to the remaining data.
- **LCMD** (Holzmüller et al., 2023) (Largest Cluster Maximum Distance): Select a batch of representative data points based on clustering.
- **BADGE** (Ash et al., 2020): Select a batch of informative and diverse data points based on clustering and the gradient magnitude of the model parameters.

*Table 2.* Significance testing results (Win/Tie/Loss counts) based on paired t-test with 0.05 significance level of `FALE` (Algorithm 1, the upper table) and `FALE`-local (Algorithm 2, the lower table) versus the rest methods. The comparison is conducted on the model performances when $10\%, 20\%, \ldots, 100\%$ of the query budget are consumed over 10 runs of experiments.

| Datasets | `FALE` versus | | | | | | | | |
| --- | --- | --- | --- | --- | --- | --- | --- | --- | --- |
| | `FALE`-local | Random | BAIT | ACS-FW | CoreSet | BADGE | LCMD | LOGO | Total |
| ct | 10/0/0 | 10/0/0 | 10/0/0 | 10/0/0 | 10/0/0 | 10/0/0 | 10/0/0 | 10/0/0 | 80/0/0 |
| diamonds | 10/0/0 | 10/0/0 | 10/0/0 | 10/0/0 | 10/0/0 | 10/0/0 | 10/0/0 | 10/0/0 | 80/0/0 |
| kegg_undir. | 10/0/0 | 10/0/0 | 10/0/0 | 10/0/0 | 10/0/0 | 10/0/0 | 10/0/0 | 10/0/0 | 80/0/0 |
| mlr_knn. | 10/0/0 | 10/0/0 | 10/0/0 | 10/0/0 | 10/0/0 | 10/0/0 | 10/0/0 | 10/0/0 | 80/0/0 |
| online_video | 2/8/0 | 10/0/0 | 10/0/0 | 10/0/0 | 2/8/0 | 2/8/0 | 2/8/0 | 10/0/0 | 48/32/0 |
| protein | 10/0/0 | 10/0/0 | 10/0/0 | 10/0/0 | 10/0/0 | 10/0/0 | 10/0/0 | 10/0/0 | 80/0/0 |
| sarcos | 10/0/0 | 10/0/0 | 10/0/0 | 10/0/0 | 10/0/0 | 10/0/0 | 10/0/0 | 10/0/0 | 80/0/0 |
| stock | 6/2/2 | 10/0/0 | 10/0/0 | 10/0/0 | 10/0/0 | 7/1/2 | 5/5/0 | 10/0/0 | 68/8/4 |
| wecs | 9/1/0 | 10/0/0 | 10/0/0 | 10/0/0 | 10/0/0 | 10/0/0 | 10/0/0 | 10/0/0 | 79/1/0 |
| Total | 77/11/2 | 90/0/0 | 90/0/0 | 90/0/0 | 82/8/0 | 79/9/2 | 77/13/0 | 90/0/0 | 675/41/4 |

| Datasets | `FALE`-local versus | | | | | | | | |
| --- | --- | --- | --- | --- | --- | --- | --- | --- | --- |
| | `FALE` | Random | BAIT | ACS-FW | CoreSet | BADGE | LCMD | LOGO | Total |
| ct | 0/0/10 | 10/0/0 | 10/0/0 | 10/0/0 | 10/0/0 | 0/10/0 | 1/9/0 | 10/0/0 | 51/19/10 |
| diamonds | 0/0/10 | 10/0/0 | 10/0/0 | 10/0/0 | 4/6/0 | 1/1/8 | 2/8/0 | 10/0/0 | 47/15/18 |
| kegg_undir. | 0/0/10 | 9/1/0 | 8/2/0 | 9/1/0 | 6/4/0 | 0/3/7 | 0/5/5 | 6/4/0 | 38/20/22 |
| mlr_knn. | 0/0/10 | 10/0/0 | 10/0/0 | 10/0/0 | 7/3/0 | 6/3/1 | 5/4/1 | 10/0/0 | 58/10/12 |
| online_video | 0/8/2 | 10/0/0 | 10/0/0 | 10/0/0 | 7/3/0 | 3/5/2 | 4/5/1 | 10/0/0 | 54/21/5 |
| protein | 0/0/10 | 10/0/0 | 10/0/0 | 10/0/0 | 10/0/0 | 9/1/0 | 10/0/0 | 10/0/0 | 69/1/10 |
| sarcos | 0/0/10 | 10/0/0 | 10/0/0 | 10/0/0 | 1/9/0 | 0/9/1 | 0/8/2 | 10/0/0 | 41/26/13 |
| stock | 2/2/6 | 10/0/0 | 10/0/0 | 10/0/0 | 10/0/0 | 2/8/0 | 1/9/0 | 10/0/0 | 55/19/6 |
| wecs | 0/1/9 | 10/0/0 | 10/0/0 | 10/0/0 | 10/0/0 | 10/0/0 | 10/0/0 | 10/0/0 | 70/1/9 |
| Total | 2/11/77 | 89/1/0 | 88/2/0 | 89/1/0 | 65/25/0 | 31/40/19 | 33/48/9 | 86/4/0 | 483/132/105 |

- LOGO (Kim et al., 2023): An FAL method that select informative and diverse data points locally.
- `FALE`: The method proposed in this paper, which performs global data selection and global model learning.
- `FALE`-local: A degenerated version of `FALE` that performs global data selection but trains the model locally. The algorithm is summarized in the appendix A.2.

Note that some of the active query strategies are originally designed for classification tasks. For these, we adopt the adaptation methods proposed by (Holzmüller et al., 2023) to extend them to regression tasks. For all the other compared methods, the global model is trained using FedAvg (McMahan et al., 2017).

We implement the regression model using PyTorch, consisting of a single layer to accommodate methods that require gradient information, as well as the existing FL framework. The model is trained by optimizing the mean squared error (MSE) with the Adam optimizer, using a learning rate of 0.01 for 25 epochs. For local active regression methods, we utilize the implementation provided by Holzmüller et al. (2023). The implementation of LOGO (Kim et al., 2023) is sourced from the authors. The FL framework is built upon the FedLab (Dun Zeng & Xu, 2021) toolbox.

## 5.2. Results

We plot the mean learning curves over 10 runs for the compared methods in Fig. 2. The mean and standard deviation of MSE on the test set are reported. Although we also include the learning curve of our proposed `FALE` method for comparison, it is important to note that `FALE` performs one-pass querying, meaning that each batch selection is independently determined and does not accumulate communication overheads.

As shown in Fig. 2, `FALE` consistently outperforms baseline methods across all benchmarks, demonstrating the effectiveness of the proposed global data selection and global model training schemes in non-i.i.d. setting. `FALE`-local often achieves the second-best performance or is comparable to the second-best method, demonstrating that our global selection strategy effectively identifies the most informative data points in each client. This allows the global model, even when aggregated using FedAvg, to maintain strong performance. The learning curves of some baselines, such as LOGO and ACS-FW, exhibit an increase in test error as more data is queried on some datasets. This phenomenon aligns with our expectations. Because there are discrepancies between the data distribution of individual clients

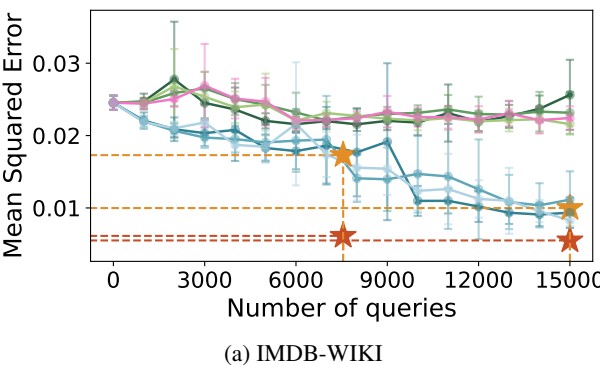 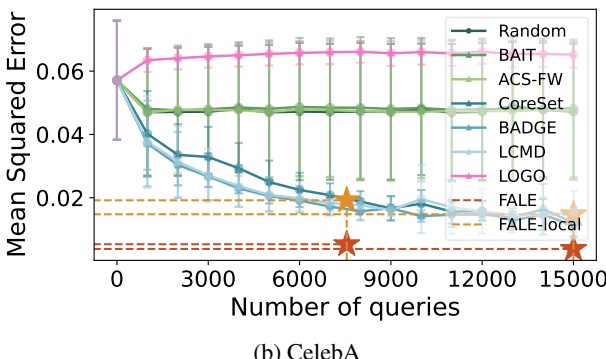

(a) IMDB-WIKI            (b) CelebA

*Figure 3.* The performance comparison results of the compared methods in the image regression datasets. The proposed methods conduct one-pass selection with query budget of 7500 and 15000.

and the latent distribution of the entire dataset. In such a challenging setting, selecting data locally based on representativeness may mislead global model learning. Among the baselines, LCMD and BADGE generally perform well, indicating that clustering-based data selection is effective in non-i.i.d. setting. This finding may serve as an inspiration for future research in this domain. The performance of Random and CoreSet varies across datasets, suggesting that there remains significant room for improvement through the development of more effective AL algorithms.

Table 2 presents the significance testing results based on a paired t-test with a 0.05 significance level, conducted over 10 runs, comparing FALE and FALE-local with the other methods. The counts of Win/Tie/Loss are calculated based on the model performances as the query budget is consumed in increments of $10\%, 20\%, \ldots, 100\%$, with 10 comparisons for each case.

The results show that FALE consistently outperforms the other baselines and rarely loses, demonstrating the effectiveness of the proposed framework. Additionally, FALE-local outperforms other local selection methods in most cases. Notably, 77 out of the total 105 losses are attributed to FALE. These findings indicate that our global selection method can effectively identify the most informative unlabeled instances in non-i.i.d. distributed data within the FL setting.

### 5.3. Study on Image Regression Dataset

We further employ 2 large scale image regression datasets to validate the effectiveness of the proposed method.

- CelebA is a facial image dataset containing 162.7K training instances and 19.9K testing instances. Following Lyu et al. (2025), we use the abscissa of the right side of the mouth as the regression target.

- IMDB-WIKI (Rothe et al., 2018) is a facial image dataset with age annotations. We adopt the dataset settings from Yang et al. (2021), which include 191.5K

images for training and 11K images for testing.

For each dataset, $5\%$ of each client's data is uniformly sampled to form the initial labeled set. We employ a pre-trained ResNet-50 to extract a 2048-dimensional feature vector for each image. All other empirical settings remain consistent with those described in the previous section.

The comparison results are presented in Fig. 3. Notably, FALE and FALE-local perform one-pass querying, with query budgets of 7,500 and 15,000, respectively. The results indicate that FALE significantly outperforms iterative selection methods, while FALE-local achieves performance comparable to the second-best method. These findings demonstrate the effectiveness and cost-efficiency in terms of both label and communication of the proposed FALE method in high-dimensional image data scenarios.

## 6. Conclusion

This paper presents a novel FAL method, FALE, which queries the most informative unlabeled data globally in a FL setting. FALE leverages FedSVD to extract cross-client query information, enabling global data selection and minimizing redundant supervision across clients. Building on this, we employ a leverage score sampling strategy for data selection and re-weighting. To fully exploit the queried data points, we further design a model training scheme that securely learns the global regression model using the masked labeled data points on the server. Theoretical analysis demonstrates that FALE requires $\mathcal{O}(d\log d)$ queries to achieve a constant factor approximation and $\Omega(d\log d + d/\epsilon)$ queries to achieve a relative error approximation with high probability. Extensive experiments on 11 regression benchmarks validate the effectiveness of the proposed method. The experimental results, compared with state-of-the-art FAL methods, show that FALE significantly reduces the number of queries to learn an effective global model on non-i.i.d. client data.

## Impact Statement

This paper presents work whose goal is to advance the field of Machine Learning. There are many potential societal consequences of our work, none which we feel must be specifically highlighted here.

## Acknowledgment

Chao Ren is supported, in part by Wallenberg-NTU Presidential Postdoctoral Fellowship, with Wallenberg AI, Autonomous Systems and Software Program (WASP); and the Swedish Research Council under contract 2024-06464. Sheng-Jun Huang is supported in part by the NSFC (U2441285, 62222605) and the Natural Science Foundation of Jiangsu Province of China (BK20222012). Lizhen Cui is supported in part by the NSFC No. 92367202. Han Yu is supported in part by the Ministry of Education, Singapore, under its Academic Research Fund Tier 1 (RG101/24); the National Research Foundation, Singapore and DSO National Laboratories under the AI Singapore Programme (AISG Award No. AISG2-RP-2020-019).

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

# A. Appendix

### A.1. Proof of Theorem 4.1

We begin by introducing the definition of subspace embedding.

**Definition A.1** ($\ell_2$ Subspace Embedding). Let $\epsilon \in (0, 1)$ be the distortion parameter. A matrix $S \in \mathbb{R}^{n_s \times n}$ is said to be an $\ell_2$ $\epsilon$-subspace-embedding matrix for $X \in \mathbb{R}^{n \times d}$ if it holds simultaneously for all vectors $\boldsymbol{\theta} \in \mathbb{R}^d$ that $(1 - \epsilon)\|X\boldsymbol{\theta}\|_2 \leq \|SX\boldsymbol{\theta}\|_2 \leq (1 + \epsilon)\|X\boldsymbol{\theta}\|_2$.

Recall that Algorithm 1 invokes FedSVD as a subroutine to securely calculate the left singular matrix of the data in each client. The clients then upload the $\ell_2$ norm of each row of the left singular matrix to the server. According to Definition 3.1, the server has the leverage scores $\boldsymbol{\tau}$ of the entire dataset $X$.

The server samples the data according to the leverage scores $\boldsymbol{p} = [p_1, \ldots, p_n]$, where $p_i = \boldsymbol{\tau}^i / \|\boldsymbol{\tau}\|_1$. Then, it calculates a re-weighting matrix $S$ where each row $i$ is equal to $\frac{1}{\sqrt{n_s p_{q_i}}} \cdot \boldsymbol{e}_{q_i}$, for $i = 1, \ldots, n_s$. It is easy to verify that $S$ is an importance sampling matrix according to Definition 3.2.

Now we introduce the following lemma.

**Lemma A.2** (Constant-factor Subspace Embedding, (Cohen et al., 2015, Theorem 7.1)). *Given $X \in \mathbb{R}^{n \times d}$. Suppose that $t_i \geq \beta \boldsymbol{\tau}^i$ for all $i \in [n]$, where $\beta \gtrsim_p \log \frac{d}{\delta}$ is a sampling parameter. Let $n_s = \sum_{i=1}^n t_i$. If $S \in \mathbb{R}^{n_s \times n}$ is a reweighted sampling matrix with sampling probability $p_i = \frac{t_i}{n_s}$ for all $i$, then $S$ is an $\ell_2$ $\frac{1}{2}$-subspace-embedding matrix for $X$ with probability at least $1 - \delta$.*

Since $\|\boldsymbol{\tau}\|_1 \leq d$, let $n_s = \mathcal{O}(d \log d)$, by applying Lemma A.2, we have that $S$ has constant-factor subspace embedding property of $X$ with probability at least $1 - \delta$.

Nota that the output of Algorithm 1 $\boldsymbol{\theta}^g$ is exactly $\bar{\boldsymbol{\theta}} = \arg\min_{\boldsymbol{\theta}} \|SX\boldsymbol{\theta} - S\boldsymbol{y}\|_2^2$. To see this, note that $\boldsymbol{\theta}^g = Q^\top \hat{\boldsymbol{\theta}}^g$ and $P$ is an orthogonal matrix which does not change the problem, i.e.,

$$\min_{\boldsymbol{\theta}} \|PSXQ\boldsymbol{\theta} - PS\boldsymbol{y}\|_2^2 = \min_{\boldsymbol{\theta}} \|SXQ\boldsymbol{\theta} - S\boldsymbol{y}\|_2^2 = \min_{\bar{\boldsymbol{\theta}}} \|SX\bar{\boldsymbol{\theta}} - S\boldsymbol{y}\|_2^2. \tag{6}$$

Next, we show that $\boldsymbol{\theta}^g$ is a constant factor approximation to the minimizer of $\min_{\boldsymbol{\theta}} \|X\boldsymbol{\theta} - \boldsymbol{y}\|_2$.

**Lemma A.3** (Constant factor approximation, (Musco et al., 2022, Theorem 3.2)). *For $X \in \mathbb{R}^{n \times d}$, $\boldsymbol{y} \in \mathbb{R}^n$. For any $\delta \in (0, 1]$, if $\bar{\boldsymbol{\theta}} = \arg\min_{\boldsymbol{\theta}} \|SX\boldsymbol{\theta} - S\boldsymbol{y}\|_2^2$, such that*

$$\|SX\bar{\boldsymbol{\theta}} - S\boldsymbol{y}\|_2 \leq (1 + \gamma) \min_{\boldsymbol{\theta}} \|SX\boldsymbol{\theta} - S\boldsymbol{y}\|_2, \tag{7}$$

*where $S \in \mathbb{R}^{n_s \times n}$ is an $\ell_2$ $\frac{1}{2}$-subspace-embedding matrix of $X$. Then, with probability at least $1 - \delta$,*

$$\|X\bar{\boldsymbol{\theta}} - \boldsymbol{y}\|_2 \leq 64(3 + \gamma)/\delta^{1/2} \min_{\boldsymbol{\theta}} \|X\boldsymbol{\theta} - \boldsymbol{y}\|_2. \tag{8}$$

By letting $\delta = 1/100$ and $\gamma = 0$, then $\|X\bar{\boldsymbol{\theta}} - \boldsymbol{y}\|_2 \leq \alpha \cdot \min \|X\boldsymbol{\theta} - \boldsymbol{y}\|_2$ for constant $\alpha$.

To achieve relative error approximation, we introduce the following lemma.

**Lemma A.4** (Relative error approximation, (Sarlós, 2006, Theorem 12 and Claim 13) ). *Given matrix $X \in \mathbb{R}^{n \times d}$. For all $1 \leq i \leq n$ set $p_i = \frac{\|\boldsymbol{e}_i^\top U\|_2^2}{\|U\|_F^2}$, where the columns in $U$ are orthonormal basis of the column space of $X$. Let $\mathcal{S} \in \mathbb{R}^{n_s \times n}$ be a row-sampling matrix such that $\Pr\left(S_{(j)} = \frac{\boldsymbol{e}_i}{\sqrt{n_s p_i}}\right) = p_i$ for all $1 \leq j \leq n_s$. For any $0 < \epsilon \leq 1$, if $n_s = \Omega(d \log d + d/\epsilon)$, it holds with probability at least $1/3$ that*

$$\|X\bar{\boldsymbol{\theta}} - \boldsymbol{y}\|_2^2 \leq (1 + \epsilon)\|X\boldsymbol{\theta}^* - \boldsymbol{y}\|_2^2, \tag{9}$$

*where $\boldsymbol{\theta}^* = \arg\min_{\boldsymbol{\theta}} \|X\boldsymbol{\theta} - \boldsymbol{y}\|_2^2$, and $\bar{\boldsymbol{\theta}} = \arg\min_{\boldsymbol{\theta}} \|SX\boldsymbol{\theta} - S\boldsymbol{y}\|_2^2$.*

Let the sampling size $n_s$ be $\Omega\left(d \log d + d/\epsilon\right)$. By applying Lemma A.4, we have

$$\|X\bar{\boldsymbol{\theta}} - \boldsymbol{y}\|_2^2 \le (1+\epsilon)\|X\boldsymbol{\theta}^* - \boldsymbol{y}\|_2^2, \tag{10}$$

holds with probability at least $1/3$, which completes the proof.

### A.2. Algorithm of `FALE`-local

We summarize the main steps of `FALE`-local method as follow

---

**Algorithm 2** `FALE`-local Algorithm

---

**Input:** query budget $n_q$, feature matrices $X_i, i = 1, \ldots, k$.
**Output:** prediction model $\boldsymbol{\theta}$
 1: **Initialization:** $\mathcal{Q}_i \leftarrow \emptyset$, for i=1,...,k.
 2: Perform FedSVD and each client obtains $U_i$, the left singular matrix that corresponds to their local data
 3: Each client $i$ calculates $\boldsymbol{\tau}_i^j = \|\boldsymbol{e}_j^\top U_i\|^2$ and sends to the server
 4: Server aggregates the leverage scores $\boldsymbol{\tau} = [\boldsymbol{\tau}_1, \ldots, \boldsymbol{\tau}_k]$ and calculates sampling probabilities $\boldsymbol{p}$, where $p_j = \boldsymbol{\tau}^j / \|\boldsymbol{\tau}\|_1$
 5: **while** Fewer than $n_q$ distinct elements sampled **do**
 6:     $q \leftarrow$ sample a number from $\{1, \ldots, \text{len}(\boldsymbol{\tau})\}$ with replacement with probability $\boldsymbol{p}$
 7:     $i \leftarrow q$ belongs to which client
 8:     append $q$ to $\mathcal{Q}_i$
 9: **end while**
10: Server sends $\mathcal{Q}_1, \ldots, \mathcal{Q}_k$ to corresponding clients
11: Clients receive the queried indexes of data and label them locally
12: Clients train a regression model locally based on their labeled data and upload the model parameters to the server
13: Server aggregates local models using FedAvg to obtain the global model $\boldsymbol{\theta}$
14: **Return:** $\boldsymbol{\theta}$

---

