# OpenReview forum: "Efficient Heterogeneity-Aware Federated Active Data Selection"
_ICML.cc/2025/Conference — ICML 2025 poster_

### Official Review · Reviewer_JwJU · 2025-03-02

**Overall Recommendation:** 1

**Summary:**

This paper considers active linear regression in the federated learning setting. It adapts the leverage score sampling to the federated learning setting for active learning. To make the method work in federated learning, it requires two components, data selection which estimates the leverage scores, and model learning which solves the linear regression problem once the data is selected and labels are obtained. For both of these two components, it applies the idea of FedSVD where it assumes that there is a trusted authority that generates random orthogonal matrices P and Q; each client to send their masked local data $PX_iQ$ to the server; the server does global computation, and send back the results to each clients.

It claims that the method is privacy preserving. It provides a theoretical analysis showing that the method achieves a similar label complexity as in the centralized learning setting. It also provides some empirical evaluation.

**Claims And Evidence:**

One of my major concerns is that I'm not convinced that the proposed method is privacy preserving. The only support for it seems to be based on the idea that "there are infinitely many possible $\bar{X}$ can be masked into X' ($PX_iQ$)" (line 182, left column), but this only guarantees that one cannot recover $X_i$ from $PX_iQ$, but recovering it up to some linear transformation is already leaking a significant amount of information in my opinion. A more precise theoretical characterization of what kind of "privacy" is preserved would be helpful.

**Essential References Not Discussed:**

N/A

**Experimental Designs Or Analyses:**

N/A

**Methods And Evaluation Criteria:**

N/A

**Other Comments Or Suggestions:**

N/A

**Other Strengths And Weaknesses:**

N/A

**Questions For Authors:**

N/A

**Relation To Broader Scientific Literature:**

N/A

**Theoretical Claims:**

I'm not very convinced that the main theoretical contribution, Theorem 4.1 is correct. It looks to me that in the proposed method, the leverage score $\tau_i$ is computed locally. Can you provide a formal proof showing that the locally computed $\tau_i$ is the same or close to the global leverage score defined as in equation (1)?

---

> ### Author Rebuttal · Authors · 2025-04-01
>
> Thank you for your careful review of our manuscript. Below we respectfully and explicitly address your main concerns point-by-point.
>
> > Q1-*Claims And Evidence*: Concerns about privacy preservation and lack of a precise theoretical characterization of what kind of "privacy" is preserved
>
> Thank you for raising this concern. We acknowledge that FALE utilizes masking techniques based on the FedSVD algorithm, which leads to partial leakage of structural information, such as the global distributional properties could be indirectly inferred from leverage scores, although direct identification of local client data remains infeasible. However, such limited structural leakage is often accepted in practical FL scenarios, please see [1-4]. Following the established FedSVD, **our method maintains the same level of data confidentiality against direct reconstruction**.
>
> To clarify this issue further, we have explicitly stated these limitations and the level of privacy protection
> provided in the revised manuscript. Specifically, **FALE adopts a one-pass querying mechanism and avoids persistent metadata tracking**. Once selected data indices are returned to clients, these mappings are discarded, reducing the risk of leakage. Furthermore, **FedSVD's privacy-preserving masking mechanism ensures that raw client data remains inaccessible**. Although the global distributional properties might be indirectly inferred from leverage scores, we wish to point that this certain degree of privacy risk is **generally acceptable** in the FL literature (see [1-4]). Hope for your understanding! We will folow and focus on more related privacy works in the future.
>
> > Q2-*Theoretical Claims*: Concerns regarding correctness and validity of Theorem 4.1, specifically the proof that locally computed leverage scores match the global leverage scores
>
> We clarify that the validity of Theorem 4.1 relies fundamentally on the correctness of leverage scores computed via FedSVD. FedSVD has been rigorously proven to securely compute exact global singular vectors from distributed, non-i.i.d. data without exposing raw client data. **Since leverage scores are directly computed from these securely aggregated global singular vectors, leverage scores computed locally by each client indeed exactly match the global leverage scores computed centrally**. Consequently, Theorem 4.1 remains rigorously correct in our federated setting.
>
> In the revised manuscript, we have rewritten Theorem 4.1 as follows.
>
> **Theorem 4.1**  *Consider FL with non-i.i.d. data. Let $k$ be the number of clients, $\epsilon \in (0, 1]$ be an error parameter, $X_i \in \mathbb{R}^{n_i \times d}$ be the corresponding data matrices, and $\mathbf{y}_i \in \mathbb{R}^{n_i}$ be the initially unknown target vector. Denote by $X = [X_1^\top, \ldots, X_k^\top]^\top$, $\mathbf{y} = [\mathbf{y}_1^\top, \dots, \mathbf{y}_k^\top]$, and $\mathbf{\theta}^\ast = \arg \min _ {\mathbf{\theta}} \| X \mathbf{\theta} - \mathbf{y} \|_2^2$.  **Algorithm 1 computes the global leverage scores for the data in each client**. Moreover, if Algorithm 1 queries $\mathcal{O}(d \log d)$ data points and outputs the model $\mathbf{\theta}^g$, then with probability at least 0.99 that:*
>
>
> $\| X \mathbf{\theta}^g - \mathbf{y} \|_2^2 \leq \alpha \| X \mathbf{\theta}^\ast - \mathbf{y} \|_2^2,$
>
>
> *with some constant $\alpha$. In addition, if it queries $\Omega(d \log d + d / \epsilon)$ data points and outputs $\mathbf{\theta}^g$, then with probability at least $1/3$ that:*
>
> $\| X \mathbf{\theta}^g - \mathbf{y} \|_2^2 \leq (1 + \epsilon) \| X \mathbf{\theta}^\ast - \mathbf{y} \|_2^2$.
>
>
>
> Finally, we would like to emphasize that FALE, to the best of our knowledge, introduces the first FAL approach with a global data selection strategy, **substantially different from existing local-selection methods and is more applicable to the practical heterogeneous FL settings**. Moreover, our theoretical analysis on query complexity represents an important and novel contribution in the FAL literature.
>
> ### References
>
> [1] Rothchild, Daniel, et al. ”Fetchsgd: Communication-efficient federated learning with sketching.” International Conference on Machine Learning. PMLR, 2020.
>
> [2] Gu, Hang, et al. ”Fedaux: An efficient framework for hybrid federated learning.” ICC 2022-IEEE International Conference on Communications. IEEE, 2022.
>
> [3] Li, Chengxi, Gang Li, and Pramod K. Varshney. ”Decentralized federated learning via mutual knowledge transfer.” IEEE Internet of Things Journal 9.2 (2021): 1136-1147.
>
> [4] Chai, D., Wang, L., Zhang, J., Yang, L., Cai, S., Chen, K., and Yang, Q. Practical lossless federated singular vector decomposition over billion-scale data. In Proceedings of the 28th ACM SIGKDD Conference on Knowledge Discovery and Data Mining, pp. 46–55, 2022.

---

> > ### Comment · Reviewer_JwJU · 2025-04-03
> >
> > Thanks for the response. My question on Theorem 4.1 is addressed.
> >
> > However, I would like to keep my score since:
> >
> > - As pointed out by Reviewer 2DKU, the main technique of this paper is a direct application of FedSVD and leverage score sampling, and there is not much new insights or new techniques. The novelty may not meet the bar for ICML.
> >
> > - I still think passing linear transform of the data to the server leaks too much information. Even for the reference [1-3] mentioned in rebuttal ([4] seems to be a simple variant of FedSVD), it looks like they're not directly passing such raw data after linear transforms.

---

> > > ### Author Response · Authors · 2025-04-04
> > >
> > > Thank you for your second-round comments and suggestions. We would like to respectfully provide clarifications on your remaining concerns.
> > >
> > > 1) Regarding novelty
> > >
> > > We respectfully argue that our work **is not merely a direct application of FedSVD and leverage score sampling to the FAL setting**. Instead, FedSVD is invoked only as a necessary sub-routine in our proposed method, enabling the secure computation of global leverage scores required for data selection. It is important to emphasize that leverage score sampling cannot be directly applied in the FL setting due to privacy concern, nor is it the only possible method suitable for FAL. With the development of FedSVD, the leverage score can be employed appropriately.
> > >
> > > Moreover, **the primary contribution of our work lies in defining and establishing an entirely new global FAL framework**, rather than simply applying existing techniques to FAL. Different with previous works, our proposed framework, including global data evaluation, global data selection, data indexing, labeling, and privacy-preserving global model aggregation, is systematically established and thoroughly analyzed **for the first time** in our work. We believe that this contribution is significant enough to meet the novelty requirement of ICML, as it clearly advances the state-of-the-art methods in FAL.
> > >
> > > Additionally, we kindly point out comparable levels of novelty and contribution from recent papers accepted at top conferences such as ICML and NeurIPS (see [1,2]). We sincerely hope you could reconsider our important contributions during your final evaluation.
> > >
> > > 2) Regarding information leakage
> > >
> > > We respectfully argue that transmitting a linearly transformed version of data reveals only rotation-invariant structural properties (e.g., singular values and eigenvalues). However, **such rotation-invariant information has already been explicitly revealed to the server at the FedSVD step**. Thus, passing linearly transformed data does not further degrade the privacy already established by FedSVD.
> > >
> > > Furthermore, we would like to emphasize clearly that our proposed method strictly adopts the secure aggregation and masking mechanisms of FedSVD [3], because we implement our method based on their code. Therefore, **the data transmission protocol and privacy guarantees in our method exactly match those proven secure in FedSVD**. Specifically, if FedSVD does not directly pass raw data after linear transformation, neither does our proposed FALE method. Thus, we believe our privacy-preserving approach remains consistent with widely accepted practices in the FL literature.
> > >
> > > Thank you once again for your feedback and for kindly reconsidering our work.
> > >
> > > [1] Zhu, Muzhi, et al. "Generative Active Learning for Long-tailed Instance Segmentation." ICML, 2024.
> > >
> > > [2] Huang, Lingxiao, et al. "Coresets for Vertical Federated Learning: Regularized Linear Regression and K-Means Clustering." Advances in Neural Information Processing Systems 35 (2022): 29566-29581.
> > >
> > > [3] Chai, D., Wang, L., Zhang, J., Yang, L., Cai, S., Chen, K., and Yang, Q. Practical lossless federated singular vector decomposition over billion-scale data. In Proceedings of the 28th ACM SIGKDD, pp. 46–55, 2022.

---

### Official Review · Reviewer_nFHH · 2025-03-03

**Overall Recommendation:** 5

**Summary:**

This paper proposes FALE algorithm to select informative data points for non-i.i.d. federated regression task. The query strategy performs global data selection using leverage score sampling, where a FedSVD technique is employed to obtain the leverage scores of all data points in federated learning. Furthermore, a global model learning scheme is proposed to fully exploit the labeled data without privacy leakage. Both theoretical and empirical studies are conducted to validate the effectiveness of the proposed method.

**Claims And Evidence:**

The paper claims that FALE enables efficient and effective cross-client data selection in federated regression. This claim is well supported by detailed analyses and a range of experiments.

**Essential References Not Discussed:**

N/A.

**Experimental Designs Or Analyses:**

I have checked the details of experimental designs, the setup is both sound and appropriate.

**Methods And Evaluation Criteria:**

The authors evaluate the approach using 11 benchmark regression datasets. The empirical FL settings align well with existing practices in the literature. The evaluation criteria include the mean MSE loss among different random data split and the learning curve of AL approaches. They are correct and appropriate.

**Other Comments Or Suggestions:**

There are some inconsistent symbols in the paper, e.g., in Algorithm 1. The authors should revise the paper again carefully.

**Other Strengths And Weaknesses:**

Strengths:
(1) The proposed method implements global active data selection, mitigating the issue of knowledge overlap among clients in federated learning. The data selection is conducted in a one-pass way, which is novel and well-motivated for FAL scenarios.
(2) A global model learning paradigm is introduced and analyzed. It exploits the encrypted labeled data in the server and learns a model that is equivalent to the centralized setting, making it suitable for privacy-preserving learning settings.
(3) The effectiveness of the proposed method is validated through comprehensive theoretical and empirical studies. The evaluation is robust. The experiments incorporate recent state-of-the-art methods and a diverse set of benchmark datasets, and the results are statistically significant.

Weaknesses:
(1) The method requires clients to encrypt and upload their local data, which might raise concerns regarding potential data leakage and increased communication overhead.
(2) The paper employs an encryption technique related to homomorphic encryption; however, it would benefit from a more thorough review of related work in this area.
(3) Experimental validation is limited to a federated learning scenario involving 10 clients. Further studies on a larger number of clients to better assess scalability.

**Questions For Authors:**

How does the communication overhead of FALE scale with an increasing number of clients and larger datasets?

**Relation To Broader Scientific Literature:**

The contributions of this paper relate to the active regression problem with $\ell_p$ loss functions.

**Theoretical Claims:**

Yes, the proofs follow the techniques in leverage score sampling. They appear to be rigorous and correct.

---

> ### Author Rebuttal · Authors · 2025-04-01
>
> Thank you very much for your thoughtful review of our paper. Below we respectfully address each of your comments point-by-point.
>
> > Q1-*Other Strengths And Weaknesses*: Concerns regarding potential data leakage and increased communication overhead
>
> Regarding data leakage concern, please see our response to reviewer JwJU for more details.
> Regarding communication overhead, we clarify that the method involves minimal overhead due to the one-pass querying strategy and efficient masking mechanisms. Specifically, the communication cost scales linearly with the number of clients and selected samples, and thus remains practical for large-scale federated scenarios.
>
> > Q2-*Other Strengths And Weaknesses*: Lack of thorough review of related homomorphic encryption works
>
> As Reviewer nFHH's suggestion, we have explicitly included and reviewed related works employing homomorphic encryption in the revised manuscript.
>
> > Q3-*Other Comments Or Suggestions*: Some inconsistent symbols
>
> We have carefully proofread the whole paper and eliminated all the grammatical, spelling, and punctuation errors. This revision enhances the readability and ensures the information is accessible without compromising the depth and accuracy of the content.
>
> > Q4-*Questions For Authors*: Clarification on how the communication overhead of FALE scales with increasing numbers of clients and larger datasets
>
> We wish to clarify further and provide detailed insights regarding communication overhead scalability explicitly as follow:
> The communication overhead of FALE involves three main components: FedSVD, data selection transmission, global model training.
>
> 1. FedSVD has been rigorously validated as highly scalable to large datasets, incurring communication costs proportional to data dimensionality rather than the total data size. Thus, this overhead remains manageable even for extremely large-scale datasets.
> 2. The cost of transmission the leverage scores and the indices of the selected data scales linearly with the total number of data instances. Each client transmits only a vector. Thus, even with an increasing number of clients or larger datasets, the incremental communication overhead remains relatively modest.
> 3. The global model training phase has a similar communication complexity with FedSVD, which shares the same scalablity to large scale datasets.
>
> Therefore, explicitly considering these three components together, **FALE scales efficiently and practically with increasing numbers of clients and larger datasets**. In the revised manuscript, we have explicitly provided quantitative analyses illustrating communication overhead of in various scenarios, thus thoroughly demonstrating its scalability and practical applicability in realistic FL settings.

---

### Official Review · Reviewer_gkkn · 2025-03-08

**Overall Recommendation:** 4

**Summary:**

This paper presents FALE (Federated Active data selection by LEverage score sampling), a novel Federated Active Learning (FAL) method for regression tasks with non - i.i.d. client data. FALE leverages FedSVD to gather global data information without exposing individual client data. For global model learning, it trains a global model on the server using masked feature matrices and label vectors of the queried data, which are then unmasked on the client side. Theoretical results are provided to validate the superiority of FALE. Experiments on 9 benchmark datasets demonstrate that FALE significantly outperforms existing SOTA methods in terms of mean squared error, validating its effectiveness.

**Claims And Evidence:**

The theoretical result is not convincing. Refer to **Theoretical Claims**.
The experiments are extensive.

**Essential References Not Discussed:**

There are no obvious essential references that were not discussed based on the information provided.

**Experimental Designs Or Analyses:**

The authors include enough baseline methods and consider sufficient benchmark datasets in experiments. The proposed FALE method outperforms other baselines on benchmark datasets.

**Methods And Evaluation Criteria:**

Please refer to **Experimental Designs Or Analyses**.

**Other Comments Or Suggestions:**

*　Many text in Fig 1 are cut by frames, please fix them.
*　Line 111, the formulation lacks an ending period.
*　Fig 2, the same label repeats 9 times, please simplify it.

**Other Strengths And Weaknesses:**

* The problem setting of federated active regression is under-explored and meaningful. Mitigating issues like client drift and imbalance that are common in federated scenarios.
* The one-pass selection mechanism and privacy-preserving model training are highly relevant for real-world federated settings.
* Limited Novelty. The proposed method is a simple application of FedSVD in the FAL field.
* Additional clarification on the computational overhead of the masking and aggregation steps would further strengthen the practical insights.

**Questions For Authors:**

The authors claim that the proposed method does not require an initially labeled dataset. However, the experiments use a labeled dataset to initialize the model. What would happen if there is no available label? Also, how does the size of the labeled set influence the performance of the proposed method?

**Relation To Broader Scientific Literature:**

The key contributions are not related to the broader scientific literature.

**Theoretical Claims:**

* The authors provide theoretical guarantees, demonstrating both constant factor approximation and relative error approximation. However, Theorem 4.1 does not seem to have any relation with the proposed FALE. How can its result demonstrate the effectiveness of FALE?
* Eq.4 seems to be meaningless. Eq.4 holds as long as $\alpha$ is large enough.
* The correctness of any proofs for theoretical claims is not checked.

---

> ### Author Rebuttal · Authors · 2025-04-01
>
> Thank you very much for your time in reviewing our paper. Below we respectfully address your concerns point-by-point.
>
> > Q1-*Theoretical Claims*: The relationship and effectiveness of Thrm. 4.1 for FALE
>
> We have improved and rewritten Thrm 4.1 to explicitly clarify its direct connection and effectiveness to FALE algorithm. Specifically, Thrm 4.1 provides a formal guarantee that the output of Algorithm 1 achieves an approximation to the optimal solution within a constant factor or relative error. This can directly verify the effectiveness of FALE, as it quantifies how close the performance of FALE is compared to the best achievable performance. Also, please refer to our response to Reviewer JwJU in Q2 for details.
>
> Regarding why these results apply in the FL context, please refer to our detailed response to Reviewer 2DKU in Q4.
>
> > Q2-*Theoretical Claims*: Meaningfulness of Eq. (4)
>
> Constant-factor approximations are standard and meaningful theoretical results widely recognized in learning theory literature (see also [1-2]). **The key significance is that the factor $\alpha$ is a fixed, bounded, and problem-independent constant rather than arbitrarily large.** This theoretical guarantee confirms that FALE’s selected samples provide sufficient global information. We have also explicitly highlighted and clarified this point in our revised manuscript.
>
> >Q3- *Other Strengths And Weaknesses*: Limited novelty
>
> We wish to clarify our contributions explicitly. To the best of our knowledge, **our proposed FALE method is the first federated active learning approach with a global data selection strategy, substantially different from existing local-selection methods.** Moreover, our theoretical analysis on query complexity represents an important contribution to FAL, demonstrating the theoretical advantages of FALE. We sincerely appreciate your feedback; in the revision, we have explicitly emphasized these novel contributions.
>
> > Q4-*Other Strengths And Weaknesses*: Clarification on computational overhead
>
> We have added the following detailed analysis in the revised paper: " **The client-side masking computation scales quadratically with the labeled samples per client ($n^L_i$) and linearly with data dimension $d$, both of which are typically modest in size.** The server-side aggregation step involves solving a regression problem, which modern computing systems can handle efficiently. Therefore, the computational overhead introduced by masking and aggregation steps is not significant, making our method practical and scalable for realistic federated scenarios."
>
> > Q5-*Other Comments Or Suggestions*: Presentation and readability issues
>
> We have carefully revised all figures and formulations (especially Fig. 1 and Fig. 2) in the paper. Specifically, text will be repositioned to avoid being cut by figure borders, repetitive labels will be simplified, and missing punctuation will be corrected. We appreciate your suggestions for improving clarity.
>
> >Q6-*Questions For Authors*: Influence of labeled set size on FALE performance
>
> We have conducted a new experiment to explore the effect of the size of the labeled dataset to the performance of FALE. The results with different initial labeled data rates (0.01, 0.03, 0.05, 0.1) of the proposed methods, after querying 1000 data points, are shown as below.
>
> | Dataset | 0.01 FALE | 0.01 FALE-local | 0.03 FALE | 0.03 FALE-local | 0.05 FALE | 0.05 FALE-local | 0.1 FALE | 0.1 FALE-local |
> |-|-|-|-|-|-|-|-|-|
> | ct   | 0.15±0.00 | 0.18±0.02 | 0.15±0.00 | 0.25±0.09 | 0.15±0.00 | 0.21±0.06 | 0.14±0.00 | 0.20±0.05 |
> | diamonds | 0.11±0.01 | 0.55±0.06 | 0.15±0.01 | 0.55±0.04 | 0.15±0.01 | 0.48±0.03 | 0.13±0.00 | 0.42±0.03 |
> | kegg_undir_uci | 0.71±0.01 | 0.74±0.01 | 0.72±0.01 | 0.73±0.01 | 0.72±0.01 | 0.73±0.01 | 0.72±0.00 | 0.72±0.01 |
> | mlr_knn_rng  | 0.51±0.01 | 0.61±0.03 | 0.51±0.00 | 0.58±0.03 | 0.50±0.00 | 0.57±0.03 | 0.50±0.00 | 0.55±0.03 |
> | online_video | 0.49±0.03 | 0.48±0.01 | 0.48±0.01 | 0.48±0.00 | 0.48±0.02 | 0.48±0.00 | 0.48±0.02 | 0.48±0.00 |
> | protein | 0.76±0.00 | 0.85±0.01 | 0.77±0.01 | 0.84±0.01 | 0.77±0.01 | 0.83±0.01 |0.77±0.00 | 0.82±0.01 |
> | sarcos | 0.16±0.01 | 0.56±0.03 | 0.18±0.01 | 0.55±0.03 | 0.17±0.01 | 0.54±0.03 |0.17±0.01 | 0.51±0.03 |
> | stock | 0.57±0.00 | 0.59±0.02 | 0.56±0.00 | 0.59±0.02 | 0.56±0.00 | 0.57±0.01 |0.56±0.00 | 0.55±0.01 |
> | wecs| 0.58±0.01 | 0.66±0.05 | 0.57±0.01 | 0.66±0.03 | 0.58±0.01 | 0.65±0.03 |0.57±0.01 | 0.61±0.03 |
>
> As shown, our proposed method is minimally affected by the size of the initially labeled dataset, demonstrating its robustness across a variety of applications.
>
> ### References
>
> [1] Woodruff, David P. "Sketching as a tool for numerical linear algebra." Found. and Trends in Theo. Comp. Sci. 10.1–2 (2014): 1-157.
>
> [2] Mahoney, Michael W. "Randomized algorithms for matrices and data." Found. and Trends in ML 3.2 (2011): 123-224.

---

### Official Review · Reviewer_2DKU · 2025-03-26

**Overall Recommendation:** 3

**Summary:**

This paper investigates the data selection problem in Federated Active Learning (FAL) and introduces FALE, a score-based sampling method. FALE leverages FedSVD to extract cross-client query information, enabling a leverage score-based sampling strategy for data selection and re-weighting. Theoretical analysis shows that FALE requires O(d log d) queries for a constant-factor approximation and Ω(d log d + d/ε) queries for a relative error approximation with high probability. Experimental results on regression benchmarks validate the effectiveness of the proposed approach.

**Claims And Evidence:**

The main contribution of this paper is a privacy-preserving data selection algorithm for regression tasks with non-i.i.d. client data in federated learning. The authors claim that the proposed method operates without requiring an initial labeled set and can select instances in a single pass, thereby reducing communication costs. Additionally, they provide a theoretical analysis of the query complexity of their approach.

A key strength of the proposed method is its inherent support for unsupervised learning, as it leverages FedSVD decomposition to extract cross-client query information. However, several concerns arise regarding the claims:

1. Efficiency and Privacy: The paper mainly references analysis of FedSVD to support its efficiency and privacy claims but does not provide an independent communication complexity or privacy analysis for FALE itself. This omission is significant because FALE introduces additional challenges beyond standard SVD tasks. A key concern is metadata tracking, where the server must maintain mappings between sample indices and client identities to return selection results. This process may introduce extra communication and storage overhead, which is not explicitly analyzed. Additionally, it poses potential privacy risks, as it could expose certain distributional properties of local datasets, thereby compromising privacy.

2. Handling of Heterogeneous Client Data: The paper claims to address federated learning with heterogeneous client data; however, its approach primarily relies on FedSVD, where masked client data is aggregated on the server into a matrix representation, effectively forming a pooled dataset. There is no explicit mechanism or dedicated analysis in the paper that directly addresses the challenges of heterogeneity in federated settings. Without specific design considerations and analysis for non-i.i.d. data, the claim that the proposed method effectively handles heterogeneous client distributions remains unsubstantiated.

Overall, while the proposed method offers an interesting approach to privacy-preserving data selection, the lack of independent analyses on communication efficiency, privacy risks, and heterogeneous data handling weakens the validity of its claims. A more thorough evaluation of these aspects would strengthen the impact and reliability of the work.

**Essential References Not Discussed:**

N/A

**Experimental Designs Or Analyses:**

The experimental design is reasonable but lacks soundness in several key aspects:

1. Scalability to Complex Models: All experiments are conducted on a single-layer model and mainly low-dimensional feature spaces. It remains unclear whether the proposed method can scale effectively to deeper architectures or more complex models in real-world applications. Testing on more diverse model architectures would provide stronger validation.

2. Federated Scalability: The experiments are limited to only 10 clients with a query budget of 5 instances per client. This small-scale setting does not reflect realistic federated learning scenarios.

3. Computational and Communication Costs: The paper claims reduced cost due to FedSVD-based selection, but there are no experiments analyzing server-side computational overhead (SVD computation) or communication cost (data collection and transmission).

**Methods And Evaluation Criteria:**

The methods and evaluation criteria presented in the paper are partially reasonable but require further refinement and justification.

Federated Active Learning and Data Heterogeneity Considerations: The approach of extracting cross-client information using FedSVD followed by leverage scoring is a reasonable choice. However, the paper does not sufficiently address the specific challenges of FAL and data heterogeneity (as discussed in the previous section). A more thorough analysis of how the method adapts to varying client distributions and the selection dynamics in an active learning scenario would strengthen its contributions.

Empirical Evaluation Scope: The experimental evaluation primarily focuses on regression performance using Mean Squared Error (MSE) as the evaluation metric. However, the paper lacks ablation studies to analyze the contribution of each algorithmic component and case studies to examine the characteristics of the selected data instances. Without these, it remains unclear whether each step of the proposed method is essential and effective, and what types of data are preferentially selected. A more comprehensive evaluation, including component-wise validation and qualitative insights, would improve the reliability and interpretability of the results.

Overall, while the proposed methodology is reasonable, a deeper investigation into federated active learning constraints and a more comprehensive empirical analysis are necessary to substantiate its claims and enhance its applicability.

**Other Comments Or Suggestions:**

N/A

**Other Strengths And Weaknesses:**

Strengths: Federated Active Learning (FAL) is a timely and practical research topic, addressing key challenges in federated learning by reducing labeling costs. The application of leverage score sampling on unlabeled data is promising, as it provides a principled way to select informative instances while preserving privacy.

Weaknesses: The paper lacks originality, as its core methodology and claims, particularly regarding privacy and communication efficiency, heavily rely on FedSVD without substantial novel extensions. While leveraging FedSVD for data selection is reasonable, the work does not introduce significant new contributions to the specific challenges of FAL, such as adaptive query strategies, uncertainty estimation, or handling of heterogeneous client data.

**Questions For Authors:**

Could you clarify key terminology such as "one-pass" (one communication between client and server?), "query budget"(numbers of instance queried?), and "query complexity" to ensure a precise understanding of their definitions and implications in the context of your method?

Based on the experiments, the query budget is set to 5 per client with 10 query rounds. Does this mean that the server alone handles the selection process, selecting and labeling 50 instances per client before returning them for training? Additionally, for trials with 15,000 queries, does the sampling algorithm converge in a way that leads to repeated selection of the same instances, potentially collapsing diversity in the selected samples?

**Relation To Broader Scientific Literature:**

N/A

**Theoretical Claims:**

The proof for Theorem 4.1 leverages FedSVD to transform distributed client data into a pooled matrix representation, applying several lemmas originally proposed for centralized matrix approximation. However, it is unclear whether these results remain valid in the non-i.i.d. and distributed setting of FL. In FL, client data distributions are typically non-i.i.d., unbalanced, and locally biased, which may violate the assumptions underlying these lemmas, many of which rely on random or structured sampling.

---

> ### Author Rebuttal · Authors · 2025-04-01
>
> We greatly appreciate your constructive suggestions. Below, we address each of your comments in detail.
>
> > Q1- *Claims And Evidence*: Independent communication complexity and privacy analysis of FALE
>
> As Reviewer 2DKU's suggestion, we have included a dedicated analysis of FALE's communication and storage complexity aspects. Specifically, in our proposed FALE method, each selected data instance is associated with two identifiers, i.e., the index of the data and client. Therefore, the communication complexity and storage overhead of transmitting leverage scores and selected data indexes are **$\mathcal{O}(n)$ and $\mathcal{O}(n_s)$**, respectively. Thus, the overhead in communication from maintaining and transmitting the metadata is **minimal and unlikely to be a practical bottleneck**, especially compared to transmission of the model gradient, which is usually much larger. Note that FALE conducts one-pass data selection, so the server **do not necessarily store the mappings** between sample indices and client identities.
>
> Regarding privacy, please refer to our detailed response to Reviewer JwJU in Q1.
>
>
> > Q2-*Claims And Evidence*: Explicit mechanism to address heterogeneity in FL settings
>
> We wish to clarify that FALE inherently addresses data heterogeneity by leveraging FedSVD, which securely computes global singular vectors on distributed and non-i.i.d. client data. Consequently, **the obtained global leverage scores accurately reflect global data structures** rather than being biased towards any specific local data distribution. This global perspective naturally addresses the challenge of non-i.i.d. and heterogeneous client data distributions.
>
> > Q3-*Methods And Evaluation Criteria*: Lack of ablation studies and  analyses of selected instances
>
> We wish to explain that the paper includes the **variant FALE-local, which can be regarded as one of the ablations to FALE**. Since it is a degenerated version of the proposed FALE method that only performs global data selection. To further enhance clarity, in the revision, we have explicitly added more detailed ablation studies and analyses exploring individual components. We have also included visual and statistical analyses of the selected data instances to better illustrate their global informativeness.
>
> > Q4-*Theoretical Claims*: Validity of Theorem 4.1 in the non-i.i.d. distributed setting
>
> We wish to clarify that Theorem 4.1 remains valid under FL settings with non-i.i.d. data distributions. Theorem 4.1 only depends on the leverage score sampling derived from the globally computed singular vectors via FedSVD. Since FedSVD has been rigorously validated to accurately compute global singular vectors securely, the leverage scores obtained exactly match those in a centralized setting. We refer to our response to Q2 of reviewer JwJU for more details and an updated version of Theorem 4.1.
>
> > Q5-*Other Strengths And Weaknesses*: Contributions on specific challenges of FAL
>
> We respectfully emphasize that, to best of our knowledge, FALE is the first FAL method to propose global data selection explicitly, departing from local query strategies common in prior works. Additionally, we provide a theoretical analysis of query complexity to provide formal guarantee, which is largely absent in existing literature of FAL. **These innovations address both heterogeneity and communication challenges** and have been highlighted in the revised manuscript.
>
> > Q6-*Questions For Authors*: Terminology clarification
>
> In the revised manuscript, we have given the clear definitions.
> **One-pass** refers to conducting the active querying only once, without multiple iterative rounds.
> **Query budget** indicates the total number of allowed instances selected for labeling in active learning.
> **Query complexity** denotes theoretical analyses quantifying how many queried instances are necessary to achieve certain performance guarantees with high probability.
>
> > Q7-*Questions*: Experimental setup clarification and concern of distribution shift}
>
> We clarify explicitly that there is a budget of labeling 50 data points (5 data querying for each client) per round for the methods that select data iteratively. FALE conducts one-pass selection in the server side. **Note that the server only handles the data selection part**, labeling always occurs locally on the client side.
>
> For your 2nd concern, active selection naturally tends to focus on the most informative instances, potentially introducing bias. However, empirical studies indicate that such a bias does not necessarily degrade performance [1].
>
> Regarding scalability, computational and communication costs, please refer to our detailed responses to Reviewer nFHH in Q4 and Reviewer gkkn in Q4.
>
>
> ### References
>
> [1] Lowell, D., Lipton, Z. C., and Wallace, B. C. (2019). ”Practical Obstacles to Deploying Active Learning.” EMNLP-IJCNLP.

---

### Decision · Program_Chairs · 2025-05-01

**Decision:**

Accept (poster)

**Comment:**

The overall quality of the paper is excellent. One major dispute is whether the proposed method can provide the desired level of privacy-preservation. In general, the federated learning framework can protect privacy in a trusted environment by embodying data locality mechanism. To provide privacy-preservaition guarantee in an untrusted envrionment, with privacy leakage and backdoor attack, a practical solution is to integrate a federated learning framework with a protection method from the cybersecurity domain.